# Microbial and Plant Derived Low Risk Pesticides Having Nematocidal Activity

**DOI:** 10.3390/toxins14120849

**Published:** 2022-12-03

**Authors:** Antonio Evidente

**Affiliations:** 1Department of Chemical Science, University of Naples Federico II, Complesso Universitario Monte S. Angelo, Via Cinthia 4, 80126 Naples, Italy; evidente@unina.it; 2Institute of Sciences of Food Production, National Research Council, Via Amendola 122/O, 70125 Bari, Italy

**Keywords:** agrarian plant disease, nematodes, natural substances, nematocidal activity

## Abstract

Microorganisms, virus, weeds, parasitic plants, insects, and nematodes are among the enemies that induce severe economic losses to agrarian production. Farmers have been forced to combat these enemies using different methods, including mechanical and agronomic strategies, since the beginning of agriculture. The development of agriculture, due to an increased request for food production, which is a consequence to the rapid and noteworthy growth of the world’s population, requires the use of more efficient methods to strongly elevate the yield production. Thus, in the last five-to-six decades, a massive and extensive use of chemicals has occurred in agriculture, resulting in heavy negative consequences, such as the increase in environmental pollution and risks for human and animal health. These problems increased with the repetition of treatments, which is due to resistance that natural enemies developed against this massive use of pesticides. There are new control strategies under investigation to develop products, namely biopesticides, with high efficacy and selectivity but based on natural products which are not toxic, and which are biodegradable in a short time. This review is focused on the microbial and plant metabolites with nematocidal activity with potential applications in suitable formulations in greenhouses and fields.

## 1. Introduction

Since ancient times, agriculture has been developed to produce food to satisfy the human needs. This request became an emergency in parallel with the increasing world population, which could reach to almost 10 billion by 2050 [1,2]. Unfortunately, this request was negatively affected by a strong reduction in natural resources, the diffused environmental pollution, and noteworthy climate changes [3,4]. In addition to these factors, farmers are forced to combat the natural enemies of agrarian plants, such as pathogens, bacteria, fungi, viruses, weeds, parasitic plants, dangerous insect, and nematodes [5,6,7]. The spread and survival of these enemies has been controlled using different methods, including mechanic and agronomy strategies. However, in the last five-to-six decades, a massive and extensive use of chemicals has occurred, with heavy consequences in terms of environmental pollution and risks to human and animal health. These problems have increased with the repetition of treatments, which are due to the resistance that enemies have developed against the pesticides used for a long time [5,8]. Thus, several multidisciplinary research groups have investigated new control strategies to develop products with high efficacy and selectivity against agrarian pests but based on natural products which are not toxic, and which are biodegradable in a short time [8,9,10,11]. Regarding the previous reviews on metabolites with nematocidal activity isolated from natural sources, SciFinder research was used to locate the article of Lorenzen and Anke (1998) [12], which describes some metabolites with insecticidal and nematocidal activities together with several natural compounds with cytotoxic, antitumoral, antiviral, and phytotoxic activities. Another review extensively describes the insecticidal activity of fungal metabolites, while only a short paragraph is dedicated to the fungal compounds with nematocidal activity. In particular, the peptides produced by *Omphalotus* spp. [13] are reported. In addition, a review only describes the fungal metabolites belonging to the azophilone family, with several different biological activities with very few compounds showing nematocidal activity [14]. Similar content is reported by Shen et al. (2015) [15], but in terms treating benzenediol lactones with a variable structures and belonging to family of fungal polyketides. Mao et al. (2014) [16] describes natural dibenzo-α-pyrones produced by fungi, mycobionts, plants, and animal feces which exhibit a variety of biological activities including nematocidal properties. Similar content was previously described by Ghisalberti (2002) [17]. Another review only describes the saponins isolated from *Medicago sativa* L., alfalfa, which is the most known plant species within the *Medicago* genus, and their nematocidal activity against different nematodes species [18]. Another review deals with extracts or secondary metabolites of the Mexican flora that showed biological activity against dangerous insects or parasitic nematodes [19]. A review about the metabolites produced by extremophilic fungi and belonging to different classes of natural compounds reported them as having different biological activities, including a nematocidal one [20]. One of the last biocontrol methods reported to combat nematodes is based on the changes in soil microbial community, the release of nematocidal compounds, and the induction of plant defenses. This strategy essentially uses biochar-based soil amendments, which is ecofriendly and compatible with a circular economy. However, as biochars induce complex and distinct modes of action, their nature and application regimes should be studied for particular pathogens and their effects must be locally checked [21]. Thus, all previous reviews only partially report about natural compounds with nematocidal activity, as some are restricted to one family of fungal or plant metabolites, some others report metabolites from different natural sources, and they also describe a lot of diverse natural compounds with different biological activities. Others extensively discuss fungal metabolites with insecticidal activity and few compounds with nematocidal activity, and some also cover a short time-span within the literature (2005–2020).

Thus, the present manuscript reports, for the first time, an overview which is focused on the microbial and plant metabolites with nematocidal activity with potential applications in suitable formulations in greenhouses and fields. The results discussed in the different sections were obtained from SciFinder research covering from 1995 to today, and these are chronologically reported in each paragraph.

## 2. Bacterial Metabolites

This section chronologically describes the source, structure, and biological activity of the bacterial metabolites, most of which showed nematocidal activity together with other interesting biological activities.

The avermectins and milbemycms are closely related 16-membered macrocyclic lactones produced by actinomycetes from the genus *Streptomyces*. *Streptomyces avermentilis* synthesized avermectins while *Streptomyces hygroscopicus* and *Streptomyces cyaneogriseus* produced milbemycms. Avermectins and milbemycins, are constituted by the identical 16-membered macrocyclic backbones which is fused with a hexahydrobenzofuran unit from C-2 to C-8a and a spiroketal unit from C-17 to C-25. The main difference diffrences between the two groups is the substituent at C-13 of macrolactone that is a disaccharaide in avermectin, whereas that position is unsubstituted in milbemycin. Avermectin B_1a_ and milbemycin D (**1** and **2**, Figure 1), are representative members of the two groups and thus compound **2** appeared to be the aglyvcone of metabolite **1**. The avermectins are produced as a mixture of eight different components designed a A_1a_, A_1b_, A_2a_, A_2b_, B_1a_, B_1b_, B_2a_, and B_2b_. The A-components have a methoxy group at the 5-position, whereas the B-components have a hydroxy group; the 1-components have a double bond between the 22- and 23-position, whereas the 2-components have a single bond with a hydroxy group at the 23-position. Finally, the a-components have a secondary butyl side chain at the 25-position, whereas the b-components have an isopropyl substituent at the 25-position. Several different alkyl substituents at C-25 can be also present in both groups. B_1_ analogues showed, followed by B2 ones, the highest toxicity and breadth of spectrum against *Haemonchus contortus*, *Ostertagia circumcincta*; *Trichostrongylus axei*; *Trichostrongylus colubriformis*; *Cooperia* spp.; *Oesophagostomum columbianu*, with a LD_50_ values in mice approximatively ranking between 15 and 50 mg/kg. Study on the mode of action of both groups using *Caenorhabditis elegans* showed that avermectins and milbemycins mediate their nematocidal activity through interaction with the common receptor, which is glutamate-gated chloride channel. Furthermore, during the same investigation, also the EC_50_ value of 89 nM was determined for milbemycin D, and a LD_50_ values 1.56 and 32.9 µg/mL were calculated for avermectin B1a based on 2 h exposure for *Meloidogyne incognita* or *Rotylenchulus reniformis* [22].

Violacein (**3**, Figure 1) is a natural violet pigment produced by several gram-negative bacteria, such as *Chromobacterium violaceum*, *Janthinobacterium lividum*, *Pseudoalteromonas tunicata* D2, *Collimonas* sp., *Duganella* sp., and *Pseudoalteromonas* spp. Compound **3** showed different potential pharmacological applications, such as in antibacterial, antitrypanocidal, anti-ulcerogenic, and anticancer drugs [23]. Violacein (**3**) was involved in oxidative stress resistance in *C. violaceum* [24]. Violacein (**3**), when produced by *J. lividum*, seemed to be involved in the natural defense of amphibians against fungal disease [25]. Furthermore, the marine bacterium *Pseudoalteromonas tunicate* produced compound **3**, which could act as an antipredator defense mechanism against protozoan grazers [26]. Other biological activities of violacein (**3**) include antioxidant, leishmanicidal, antifungal, and antiviral activities. Violacein (**3**) showed nematocidal activity against *Caenorhabditis elegans* with LC_50_ > 30 mM. In addition, the bacterial accumulation in the nematode intestine, which determined tissue damage and apoptosis, was induced by compound **3** and using *Escherichia coli*. This process occur also in nematodes, such as *C. elegans*, which activate a well-defined innate immune system to defend against pathogens. This defense mechanism, studied with compound **3** isolated from marine bacterium *Microbulbifer* sp. D250, employs the DAF-2/DAF-16 insulin/IGF-1 signaling (IIS) component to modulate sensitivity to violacein-mediated killing. On the basis of these results, violacein appeared to be a potential antinematode [27].

Here, 4-Oxabicyclo[3.2.2]nona-1(7), 5,8-triene was isolated together with the well-known compounds, such as (3*S*,8a*S*)-hexahydro-3-methylpyrro[1,2-a]pyrazine-1,4-dione, phenylacetamide, cyclo(L-Pro-L-Val), lauric acid, and methyl elaidate (**4**–**9**, Figure 1), from the culture filtrates of *Bacillus* strain SMrs28, which was obtained from the rhizosphere soil of the toxic plant *Stellera chamaejasme* [28]. In preliminary tests, the bacterial culture filtrates showed strong nematocidal activity against the pinewood nematode *Bursaphelenchus xylophilus*, which is a major threat to forestry leading to a billion US dollars of economic losses per year, and the tuber rot nematode *Ditylenchus destructor* [29]. In fact, compounds **4**–**6** showed toxicity against both nematode *B. xylophilus* and *D. destructor*, with a LC_50_ values of 904.12, 451.26, and 232.98 μg/mL and 1594.0, 366.62, and 206.38 μg/mL at 72 h, respectively [29].

Prodigiosin (**10**, Figure 1) is a red pigment produced from a few bacteria species, such as *Serratia*, *Pseudomonas*, and *Streptomyces*. Its isolation and structural characterization was first reported when compound **10** was obtained from massive culture filtrates of *Serratia* sp. KH-95b [30]. Prodigiosin (**10**), isolated from *Serratia marcescens*, was tested against nematodes at their juvenile stage and showed an effect against juvenile stages of *Radopholus similis* and *Meloidogyne javanica* at low concentrations (LC_50_ values, 83 and 79 μg/mL, respectively) as compared with the positive control of copper sulphate (LC_50_ values, 380 and 280 μg/mL, respectively). The pigment also exhibited inhibition on nematode egg-hatching ability [31].

Thumolycin (**11**, Figure 1), which is a lipopeptide, was isolated from *Bacillus thuringiensis*, which is a bacterium widely used as a bio-insecticide. Thumolycin has a molecular weight of 696.51 Da and a predicted molecular formula of C_38_H_64_N_8_O_4_, but its structure has not yet been determined. Compound **11** showed a broad spectrum of antimicrobial and nematocidal activities. In particular, *C. elegans* was significantly inhibited by thumolycin (**11**), which, when tested at a higher concentration, induced a higher mortality in this nematode. When *C. elegans* was treated with 600 U of thumolycin, less than 10% nematodes survived. These experiments indicated that thumolycin has inhibitory activity against a wide range of bacteria and effective nematocidal activity [32].

Aureothin and alloaureothin (**12** and **13**, Figure 1) were isolated from an endophytic bacterium strain, which was identified as *Streptomyces* sp. AE170020, and appeared to be a rich source of bioactive secondary metabolites with potential as environmentally benign agents. In fact, both metabolites **12** and **13** showed a significant nematocidal activity suppressing the growth, reproduction, and behavior of *B. xylophilus*. Pine trees are one of the most important forest plants in ecosystems, as they are widespread in natural reserves, parks, and urban ornamental landscapes, and are also source of wood with high economic value for different practical uses. Unfortunately, pine is affected by different pests, such as the pathogen fungus *Sphaeropsis sanipea*, a producer of phytotoxins [33], and by pine wood nematode (PWN). The *B. xylophilus* PWN is the causal agent of most serious pine wilt diseases. In in vivo experiments, extracts of a strain of *Streptomyces* sp. AE170020 significantly suppressed the development of pine wilt disease in 4-year-old plants of *Pinus densifora*. The LC_50_ values of aureothin (**12**) on different life stages of *B. xylophilus* (J2s, J3s, and J4s/adults) were 0.81, 1.15, and 1.54 μg/mL, respectively, while the values of alloaureothin (**13**) were 0.83, 1.10, and 1.47 μg/mL, respectively. Compared with the positive control abamectin, both compounds showed higher nematocidal activity against *B. xylophilus* at all tested life stages, and exhibited similar mortality rates. Thus, compounds **12** and **13** represent an important tool to combat *B. xylophilus*, and could be proposed for a suitable bioformulation to develop a natural nematicide [34]

## 3. Fungal Metabolites

This section chronologically reports the source, structure, and biological activity of the fungal metabolites, most of which showed nematocidal activity together with other interesting biological activities.

Some studies were carried out on five *Arthropotrys* strains to examine their ability to produce metabolites with nematocidal activity. In fact, the well-known linoleic acid was isolated from *Arthrobotrys conoides* and *Arthrobotrys oligospora* [35]. A lot of fungi belonging to *Ascomycetes Pyrenomycetes* and *Discomycetes* genera were the object of a similar investigation. A total of 29 isolates belonging to 18 genera produced metabolites with nematocidal activity against *C. elegans*, out of a total of 267 extracts of culture filtrates of the different stains examined [35]. Here, 1-Methoxy-8-hydroxynaphthalene, 1,8-dimethoxynaphthalene, and 5-hydroxy-2-methyl chromanone (**14**–**16**, Figure 1) were isolated from *Daldinia concentrica* [36]. Both naphthalene derivatives (**14** and **15**) showed nematocidal activity against *C. elegans* with LD_50_ values of 10 and 25 μg/mL, respectively, in addition to cytotoxic and antimicrobial effects, while the chromanone **16** had no nematocidal activity [35].

Here, 14-*Epi*-dihydrocochlioquinone B and 14-*epi*-cochlioquinone B (**17** and **18**, Figure 1) were isolated from *Neobulgaria pura* [37] and showed nematocidal activity towards *C. elegans* and *Meloidogyne incognita* [35]. Furthermore, the close cochlioquinone A (**19**, Figure 1), which was produced by a *Helminthosporium* species, competed for the ivermectin binding site on the membrane receptor in nematodes [38]. Ivermectin is the didroderivative of averctim, which as abovereported was originally isolated from soil in Japan as a part of a collaborative program to select microorganisms on the basis of novel microbiological characteristics and a wide variety of pharmacological and chemotherapeutic assays [39].

Here, 5-Pentyl-2-furaldehyde (**21**, Figure 1), was isolated as a metabolite with nematocidal activity from the culture filtrates of an unidentified species of the Dermateaceae family. The strain was collected in Australia. Compound **21** was also isolated from *Irpex lacteus*, which is a wood-inhabiting basidiomycete [40], and showed moderate activity against *Aphelenchoides besseyi*, *M. incognita*, and *C. elegans* with LD_50_ values 60 of and 75 μg/mL when tested on the penultimate and last nematode, respectively [40].

Lachnumon (I), lachnumol A, mycorrhizin A, chloromycorrhizin A, and (1’-*E*)-dechloromycorrhizin A and its *Z*-isomer (**22**–**27**, Figure 2) were isolated from *Lachnum papyraceum*, which was collected in southern Germany. All the metabolites showed nematocidal activity against *C. elegans* but appeared almost inactive towards *M. incognita* [41,42]. The LD_50_ values recorded for compounds **22**–**27** were 25, 5, 1, 100, 2, and 2 μg/mL, respectively [35]. Furthermore, the effect the of substitution of chlorine, which is present in compounds **22**–**25,** with bromine was investigated by growing the fungus in a medium containing CaC1_2_ with an excess of CaBr_2_. From the organic extract of these culture filtrates, lachnumol A and mycorrhizin A (**23** and **24**) were isolated in traces while the other were absent. Furthermore, six derivatives of mellein were highlighted by analytical HPLC [42,43]. They were identified as 6-hydroxymellein, 4-chloro-6-hydroxymellein, 6-methoxymellein, 4-bromo-6-hydroxmellein, 4-chloro-6-methoxymellein 6,7-dihydroxymellein, and 4-chloro-6,7-dihydroxymellein (**28**–**34**, Figure 2). All the mellein analogues showed no nematocidal activity except the moderate effect exhibited by compound **34** against *C. elegans* [35]. Hydroxymellein (**28**), which was previously isolated together with mycorrhizin A (**24**) and gilmicolin from cultures of *Gilmaniella humicola*, was proposed to be a biosynthetic precursor of mycorrhizin A and gilmicolin [44,45]. On the basis of these results, the hypothesized steps leading from 6-hydroxymellein to lachnumols and mycorrhizins seemed almost blocked, and it seemed that 6-hydroxymellein, which accumulates, could be transformed into derivatives **28**–**34**. Here, 4-Chloro- and 4-bromo-derivatives (**29** and **31**) were also synthesized. In another experiment, CaBr_2_ was later added to the medium, and other different metabolites were produced as bromo-containing analogues. They were lachnumons B1 and B2 (**39** and **39**, Figure 2) and mycorrhizins B1 and B2 (**40** and **41**, Figure 2) [41,46,47], together with mycorrhizinol (**46**, Figure 2), which was previously isolated from *Gilmaniella humicola* [43]. In addition, other metabolites were isolated, such as papyracons A-G (**35**–**37** and **42**–**45**, Figure 2) and metabolite **47** (Figure 2). The last compound (**47**), which is trisubstituted 4*H*-sprobenzofuran-7-(5*H*)-one, was identified as (1′*S*,2′*R*,5*S*)-5-hydroxy-2′-((*R*)-1-hydroxyethyl)-2-methyl-4*H*-spiro[benzofuran-6,1′-cyclopropan]-7(5*H*)-one. All five bromo-analogues (**31** and **38**–**41**) bear the bromine in the side chain. The brominated and the chlorinated compounds had very similar biological activities however, brominated lachnumons (**38** and **39**) or mycorrhizins (**40** and **41**) were slightly less active than their corresponding chlorinated analogues. All rnycorrhizins exhibited high activities against *C. elegans* with LD_50_ values recorded for compounds **38**–**41**, **46** and **35**–**37**, and **42**–**45** of 50, 25, 2 and 5, 100, and 25, 50, and 50, and 50, 50, 50, and 50 μg/mL, respectively, but are practically inactive against *M. incognita* [35].

Omphalotin (**48**, Figure 2), which is a cyclic peptide, was isolated together with sesquiterpene illudin M and the 3-(3-indolyl)-*N*-methylpropanamide (**49** and **50**, Figure 2) from the cultures of *Omphalotus olearius*. This fungus is a wood-inhabiting basidiomycete that spreads worldwide, particularly on olive, oak, and chestnut trees. Omphalotin (**48**) tested against *M. incognita* and *C. elegans* was more active than nematicide ivermectin, which is now commercialized, with LD_50_ 18.95 and 0.57 μg/mL values. Compounds **49** and **50** were inactive against both *nematode* up to 100 μg/mL [48].

A metabolite, named MK7924 (**51**, Figure 3), with nematocidal activity, was isolated from the culture filtrates of *Coronophora gregaria* L2495. Here, MK7924 (**51**), which is a highly methylated polyketide bearing two mannose residues, was related to the antibiotics TMC-151s [49,50], TMC154s [51], TMC-171s [51], and roselipins which are inhibitors of diacylglycerol acyltransferase [52,53,54]. Although compound **51** did not show antibacterial activity against *B. subtilis*, *S. aureus*, *E coli*, or *P. aeruginosa*, it exhibited antifungal activity against *Aspergillus niger* and nematocidal activity against *C. elegans* [55].

Four naphthalenones, namely 4,8-dihydroxy-3,4-dihydronaphthalen-1(2*H*)-one, 4,5-dihydroxy-3,4-dihydronaphthalen-1(2*H*)-one, 4,6,8-trihydroxy-3,4-dihydronaphtha- len-1(2*H*)-one), and 3,4,6,8-tetrahydroxy-3,4-dihydronaphthalen-1(2*H*)-one, also known as *cis*-4-hydroxyscytalone, (**52**–**55**, Figure 3) were isolated from the culture filtrates of *Caryospora callicarpa* [56]. The fungus was collected from a freshwater habitat in Yunnan Province, China. All four metabolites showed noteworthy nematocidal activity against *B. xylophilus*, which is both a plant-parasitic and fungal-feeding nematode that causes multimillion dollar loses to pine forests, especially in some Asian countries [57,58]. Compound **55** was more active than compound **52** and was, in turn, more active than metabolite **54**. The least toxic appeared to be compound **53**. Furthermore, compound **55**, having a 3-hydroxy group, was more active than the other compounds. Compounds **53** and **54**, bearing a 5- and a 6-hydroxy group, respectively, showed a tendency to reduce activity. The activity of metabolites **52**–**55** significantly increased with the length of the exposure times at the same concentration. In fact, they showed higher antinematodal activity against *B. xylophilus* in evaluations at 36 h than at 12 and 24 h exposure [56]. The LD_50_ values, depending on the three different exposure times (12, 24, and 36 h) were 540.2, 436.6, and 209.0 μg/mL for compound **52**, 1169.8, 461.3, and 229.6 μg/mL for compound **53**, 1011.6, 522.5, and 220.3 μg/mL for compound **54**, and 854, 468, and 206.1 μg/mL for compound **55**. These results suggested that the action modes of these compounds were systemic, rather than being contact poisons or anti-feedants [59].

Five preussomerin analogues, named ymf 1029A A, B, C, D, and E (**56**–**60**, Figure 3), were isolated together with the known preussomerin C, preussomerin D, (4*RS*)-4,8-dihydroxy-3,4-dihydronaphthalen-1(2*H*)-one, and 4,6,8-trihydroxy-3,4-dihydronaph- thalen-1(2*H*)-one (**61**–**64**, Figure 3), from the liquid cultures of an unidentified freshwater fungus YMF 1.01029. This fungus was obtained from the split of decaying branches of an unidentified tree near Lake Fuxian in Yunnan Province, China. All the isolated compounds were tested against *B. xylophilus*, showing weak nematocidal activity, with IC_50_ values between 100 and 200 µg/mL at the 24 h time point. Among compounds **56**–**64**, preussomerin D (**62**) exhibited the most potent toxicity, while the two naphthalenones, metabolites **63** and **64**, showed the weakest activity. All compounds had weaker activity when compared with the commercial nematicide avermectin. The *bis*-spirobisnaphthalene pharmacophore appears to be an important structural feature to impart high activity, because all tested *bis*-spirobisnaphthalene metabolites showed a stronger nematocidal activity than naphthalenone **63** and **64** [60].

Thermolides A–F (**65**–**70**, Figure 3), which belong to a class of PKSNRPS hybrid metabolites constituted by a 13-membered lactam-bearing macrolactone, were isolated from a thermophilic fungus *Talaromyces thermophilus*. Macrocyclic PKS-NRPS hybrid metabolites are a unique family of natural products, essentially produced by bacteria with broad and outstanding biological activities. All the metabolites **65**–**68** were assayed against three types of nematodes, including the root-knot nematode *M. incognita*, pine-wood nematode *B. siylopilus*, and free-living nematode *Panagrellus redivevus* [61]. Compounds **65** and **66** showed the strongest activities against all the worms, with LC_50_ values ranging from 0.5–1.0 µg/mL, similar to those of the avermectin used as control, while compound **67** and **68** had, respectively, moderate and weak inhibitory effect on the same organisms [62]. The gene ThmABCE from this fungus is fundamental for thermolide synthesis. Furthermore, a heterologous and engineered expression of the Thm genes in *Aspergillus nidulans* and *E. coli* induced a strongly increased yield not only in thermolide production, but also in that of different esterified analogues, such as butyryl- (thermolides J and K) hexanoyl-, and octanyl-derivatives or mixed thermolides. In addition, thermolides L and M were also obtained via genome mining-based combinatorial biosynthesis, and represent the first L-phenylalanine-based thermolides [63].

Palmariol B, 4-hydroxymellein, alternariol 9-methyl ether, and botrallin (**71**–**74**, Figure 4) were isolated from the endophytic fungus *Hyalodendriella* sp. All the compounds were assayed for antibacterial, antifungal, antinematodal, and acetylcholinesterase inhibitory activities. The antimicrobial activity was tested against bacteria, such as *B. subtilis*, *Pseudomonas lachrymans*, *Ralstonia solanacearum*, and *Xanthomonas vesicatoria*, and against the fungus *Magnaporthe oryzae*. Compounds **71**–**74** showed activity against *C. elegans* with IC_50_ values of 56.21, 86.86, 93.99, and 84.51 μg/mL, respectively. Here, 4-Hydroxymellein (**72**) had the strongest antibacterial activity. Palmariol B (**71**) showed stronger antimicrobial, antinematodal, and acetylcholinesterase inhibitory activities than alternariol 9-methyl ether (**73**). This last result suggested that the chlorine substitution at position 2 may be a structural feature important for bioactivity [64].

Gymnoascole acetate (**75**, Figure 4) was isolated from *Gymnoascus reessii* za-130, which was obtained from the rhizosphere of tomato plants infected by the root-knot nematode *M. incognita*. Gymnoascole acetate (**75**) showed strong toxicity against *M. incognita* second-stage juvenile (J2) viability, while exposure to its solution of 36 μg/mL for 24 h determined 100% paralysis of J2 stage (EC_50_ = 47.5 μg/mL) [65]. Chaetoglobosin A and its derivate 19-*O*-acetylchaetoglobosin A (**76** and **77**, Figure 4, which were produced by *Ijuhya vitellina* (Ascomycota, Hypocreales, Bionectriaceae), found in wheat fields in Turkey, showed significant nematocidal activity against *Heterodera filipjevi*, which was paralyzed when both compounds were tested at 50 and 100 μg/mL. In addition, at 300 μg/mL, chaetoglobosin A had higher toxic effects and caused nematode mortality of *I. vitellina* destructively-parasitizing eggs inside cysts of nematode. The parasitism was also reproduced in in vitro studies [66].

Grammicin (**78**, Figure 4), which is a dihydrofuranone, was isolated from *Xylaria grammica* KCTC 13121BP, showing a strong nematocidal activity against *M. incognita*. The fungus was isolated from a lichen, *Menegazzia* sp., which was collected on Giri Mountain in Korea [67]. Compound **78**, which was also previously isolated from the same fungus collected from wood in Cameroon and Peru [68], is a structural isomer of the well-known mycotoxin patulin (**79**, Figure 4). The latter compound (**79**) was first isolated in 1943 from *Penicillium griseofulvum* and *Penicillium expansum* [69] and then, as recently reviewed [70], from several species belonging to not less 30 genera including *Penicillium*, *Aspergillus*, *Paecilomyces*, and *Byssochlamys*. Compound **79** is the most common mycotoxin found in apples and apple-derived products and other food, and is associated with immunological, neurological and gastrointestinal outcomes with high human health risks [71]. Grammicin (**78**) showed strong nematocidal activity against *M. incognita* in J2 juvenile mortality and eggs-hatching inhibition with EC_50_ values of 15.95 and 5.87 μg/mL, respectively, compared to *trans*-cinnamaldehyde used as positive control, which showed in both assay EC_50_ values of 18.34 and 10.50 μg/mL, respectively. The same compound exhibited weak antibacterial effects against several microorganisms responsible for severe crop diseases [67]. Furthermore, it exhibited very low or no cytotoxic activity when assayed against a human first-trimester trophoblast cell line SW.71. Instead, patulin (**79**), in the same bioassays, showed a weak nematocidal EC_50_/72 h value of 115.67 μg/mL and strong antibacterial and cytotoxic activities. In addition, compared with *trans*-cinnamaldehyde, grammicin (**78**) showed comparable J2 killing activity but a stronger egg-hatching inhibitory effect. These results suggest that grammicin and its fungal producer have potential for biocontrol of root-knot nematode disease in crops [67].

Seven highly oxygenated and differently functionalized spirodioxynaphthalene, named sparticolins A–G (**80**–**86**, Figure 4) were isolated from the new species of Dothideomycetes, the ex-type strain of *Sparticola junci*, which was introduced as a member of the family Sporormiaceae [72]. These fungi are mostly saprobic on dung, but sometimes occur on other substrates, including plant debris, soil, and wood [73]. Sparticolins A–E (**80**–**84**) showed only weak toxicity against the nematode *C. elegans*, while sparticolin F (**85**) showed a moderate nematocidal activity, and sparticolin G (**86**) was not tested due to there being an insufficient amount. The LD_50_ values recorded for compounds **80**–**85** were as follows: 50, 50, 25, 50, 50, and 12.5 μg/mL, respectively. Sparticolin B (**81**) inhibited the gram-positive bacteria *B. subtilis*, *Micrococcus luteus*, and *Staphylococcus aureus*, while sparticolin G (**86**) showed antifungal activities against *Schizosaccharomyces pombe* and *Mucor hiemalis*. The latter two compounds (**81** and **86**) also showed moderate cytotoxicity against seven mammalian cell lines [73].

Six sesquiterpenes (**87**–**79**, Figure 4) and cyclodepsipeptides (**93**–**95**, Figure 4) were isolated from *Trichoderma longibrachiatum*, which is a fungus obtained from the root of *Suaeda glauca*, a highly halophilic plant collected from the intertidal zone of Jiaozhou Bay, Qingdao, China. Compound **87**, which possess a rare an original norsesquiterpene tricyclic-6/5/5-[4.3.1.0^1,6^]decane skeleton, was named trichodermene. Compound (**87**) and the sesquiterpenes **88** and **89** showed significant antifungal activities against *Colletotrichum lagenarium*, even better than those of the commercial synthetic fungicide carbendazim. A similar activity was exhibited from the same compounds against carbendazim-resistant *Botrytis cinerea*. The sesquiterpenes **91** and **92** showed nematocidal activity when assayed at 200 μg/mL at J2s lethal rate of *M. incognita* of 38.2 and 42.7%, respectively. Cyclodepsipeptides **93**–**95** showed moderate nematocidal activities against the southern root-knot of the same nematode *M. incognita* with IC_50_ values of 149.2, 140.6, and 198.7 μg/mL, respectively [74].

## 4. Plant Metabolites

This section chronologically describes the source, structure, and biological activity of plant metabolites, most of which showed nematocidal activity together with other interesting biological activities.

Twenty-four secondary metabolites were isolated from *Bupleurum salicifolium* [75], which is a plant native to the western Canary Islands from Gran Canaria to El Hierro, where it is frequently found up to 1000 m above sea level [76]. The plant is highly specialized in biosynthesizing secondary metabolites, principally lignans, coumarins, and flavonols, which all derive from shikimic acid and belong to different classes of natural compounds (Dewick, 2002) [77]. All the metabolites were tested against viruses, gram-positive and gram-negative bacteria, the yeast *Candida albicans*, the nematodes *G. pallida* and *G. rostochiensis*, the insect *Spodoptera littoralis*, and the crustacean *Artemia salina*. These compounds were also tested against tumoral and non-tumoral cell lines. In particular, considering the limited amount available, only dibenzyl-butyrolactone, lignans, such as guayarol, buplerol, matairesinol and its dimethyl ether, bursehernin, pliviatolide, thujaplicatin, methyl ether (**96**–**102**, Figure 5) and 2-chloro-matairesinol, nortrachelogenin, nortrachelogenin triacetate, and 2-hydroxy-thujaplicatin-methyl ether (**103**–**106**, Figure 5) were tested on potato cyst nematode hatching using *G. pallida* and *G. rostochiensis*. After 14 days, all the compounds assayed stimulated the hatching of more juveniles than distilled water (negative control). In particular, matairesinol and bursehernin (**98** and **100**) significantly reduced hatching by 70% and 55%, respectively, when compared to the positive control agent. The HID recorded for bursehernin (**100**) was 16.42 μg/mL, while no differences were observed in the inhibition of *G. pallida* or *G. rostochiensis*. These results suggested that the presence of a methylene-dioxy group in the aromatic ring B of the dibenzyl-butyrolactone skeleton plays a significant role to impart nematostatic activity. When the methylene-dioxy group was substituted by a methoxy and a hydroxyl group, as in buplerol (**97**), or two hydroxy groups, as in guayarol (**96**), a significant reduction in the activity was observed. Furthermore, in compounds lacking the methylenedioxy group, the activity increased according to the number of free hydroxy groups present, as observed in compounds **96** > **98** > **97** > **105**. Nortrachelogenin triacetate (**105**) bears an acetyl group at position 2 in the lactone ring, which could be a consistent steric hindrance between this compound and the receptor on the nematode eggshell whose existence was hypothesized by Atkinson and Taylor (1980; 1983) [78,79]. None of the compounds tested showed nematocidal activity when tested on second-stage juveniles of *G. pallida* and *G. rostochiensis* [75]. Successively, some of the same authors tested 22 aromatic derivatives and the conjugated carbonyl compound *t*-3-penten-2-one for nematocidal activity against the same 2 nematodes, namely *G. pallida* and *G. rostochiensis*. Among all the compounds assayed, nine showed high toxicity on the infective stages (second instar juveniles) of the nematodes, with a LC_50_ ranking from 2 × 10^−6^ to 1.26 × 10^−3^ M. As expected, the toxicity is due to the presence of a conjugated carbonyl system [80].

The cyclic hydroxamic acids are common secondary metabolites found in plants of the Poaceae family, such as corn, wheat, and rye, and known for the allelopathy of rye (*Secale cereale*). The latter plant is well-known for its allelopathic activity. Some of cyclic hydroxamic acids, such as DIBOA (2,4-dihydroxy-(2*H*)-1,4-benzoxazin-3(4*H*)-one), DIMBOA (2,4-hydroxy-7-methoxy-(2*H*)-1,4-benzoxazin-3(4*H*)-one) (**107** and **108**, Figure 5), and their degradation products BOA (benzoxazolin-2(3*H*)-one) and MBOA (6-methoxy-benzoxazolin-2(3*H*)-one) (**109** and **110**, Figure 5) are commercially available and, thus, were used to test their toxicity against *M. incognita* second-stage juveniles (J2) and eggs and mixed-stages of *Xiphinema americanum* (*X. americanum*). The LC_50_ value of 74.3 μg/mL for DIBOA was recorded when assayed against *M. incognita* eggs after 168 h exposure, while there was no possible recorded any value for the other compounds. In the assay on *M. incognita* J2 mortality, the LD_50_ values were 20.9, 46.1, and 49.2 μg/mL for compound **107**, **108**, and **110**, respectively. For compound **108**, no value was measured. In the assay against *X. americanum*, the LD_50_ values recorded after 24 h of exposure were 18.4 and 48.3 μg/mL for compounds **107** and **108**, respectively, while compounds **109** and **110** had no effect on nematode mortality. These results showed that *X. americanum* was more sensitive to DIBOA and DIMBOA (**107** and **108**) than *M. incognita* J2, while eggs of *M. incognita* were less sensitive to the hydroxamic acids than J2. Only DIBOA (**107**) resulted in a 50% reduction in egg hatching; MBOA (**110**) was not toxic to *X. americanum* or *M. incognita* eggs but was toxic to *M. incognita* J2. Furthermore, BOA (**109**) was the least toxic hydroxamic acid tested and did not reduce *M. incognita* egg hatching after 1 week of exposure or increase *X. americanum* mortality after 24 h of exposure. These results showed that the presence of 4-hydroxy-2*H*-1,4-oxazin-3(4*H*)-one is determinant of the imparted toxicity, as this activity was strongly reduced or lost when this residue was substituted by oxazol-2(3*H*)-one [81].

Coumarins are a large group of naturally occurring compounds widely distributed in the Apiaceae, Rutacea, Asteraceae, and Fabaceae plant families, and are known for their phytotoxic, fungitoxic, insecticidal, antibiotic, and nematocidal activity [82]. Furanocoumarins showed nematocidal activity. In particular, 8-geranylpsoralen, imperatorin, and heraclenin (**111**–**113**, Figure 5) exhibited nematocidal activity against *B. xylophilus* and *Panegrellus redivivus*. The LD_50_ values recorded for compounds **111**–**113** after 72 h of exposure were 188.3, 161.7, and 114.7 μg/mL and 117.5, 179.0, and 184.7 μg/mL when assayed against *B. xylophilus* and *P. redivivus*, respectively [82].

Ruixianglangdusu B, umbelliferone, chamaejasmenin C, daphnoretin 7-methoxyneochaejasmin A, (+)-chamaejasmine, chamaechromone, and isosikokianin A (**114**–**120**, Figure 5) were isolated from the organic extract of *Stellera chamaejasme* L. roots, which showed significant nematocidal activity against *B. xylophilus* and *Bursaphelenchus mucronatus* [83]. The eight metabolites were tested against J2s of *B. xylophilus* and *B. mucronatus*. The LC_50_ values recorded for compound **114**–**121** depending on the exposure time (24, 48, and 72 h) when assayed against *B. xylophilus* were as follows: 227.4, 71.6, and 15.7 μM; 1.3 × 10^7^, 5.7 × 10^7^ and 3.3 μM; 47.8, 3.1 and 2.7 μM; 1.7 × 10^4^, 1.1 × 10^4^ and 65.3 μM; <0.001, 3.4 and 167.3 μM; 16.5, 8.8, and 4.7 μM; 0.7, 10.3 and 36.7 μM; 147.7, 385.2 and 2.2 × 10^2^. Similarly, the LC_50_ values recorded at exposure of 12, 24 and 72 h for compound **114**–**121** when assayed against *B. mucronatus* were as follows: 1.8 × 10^3^, 160.2 and 0.6 μM; 2.6 × 10^3^, 851 and 33.4 μM; 2.3 × 10^6^, 169.9 and 3.1 μM; 463.5, 156.7 and 0.05 μM; 1.8 × 10^4^, 384.2 and 151.1 μM; 1.8 × 10^3^, 1.6 × 10^3^, 5.1 × 10^3^ μM; 327, 5.7, 0.003 μM; 2.6 × 10^4^, 32.5 and 2.3 μM. These results showed that chamaejasmenin C (**117**) and (+)-chamaejasmin (**119**) showed significant toxicity against *B. xylophilus* at a concentration of 100 µM at 72 h. Umbelliferone (**115**), daphnoretin (**116**) and chamaechromone (**120**) exhibited moderate activity at a concentration of 800 µM at 72 h, while compound **114**, **118** and **121** showed weak activity when tested in the same conditions. The nematocidal activities of the eight purified compounds against *B. mucronatus* were similarly observed at 72 h after treatment but the toxicity values of compounds **117** and **120** were highest at a concentration of 400 µM. The nematocidal activity of compound **119** was strongest against *B. mucronatus* at the lowest test concentration, while the most toxic compounds were **114**, **116**, and **120**, with LC_50_ values ranging from 0.003 to 0.6 µM, which were comparable with that of the lambda cyhalothrin (LC_50_ = 1.1 µM) used as the positive commercial control [83].

Medium-chain fatty acids and phenolic acids were the main component of the organic extract of *Picria fel-terrae*. The plant extract showed toxicity against free-living nematode *C. elegans* and the parasitic nematode *Haemonchus contortus*, killing *C. elegans* adults and inhibiting the motility of 48 exsheathed L3 of *H. contortus*. The same extract had minimal cytotoxic activity in mammalian cell 49 culture [84].

A screening was carried out carried out for 790 plant metabolites, including those obtained from *Tagetes* spp., *Azadirachta indica*, and *Capsicum frutescens*, and involved testing their nematocidal activity against *C. elegans*. A total of 10 compounds proved to be toxic, 3 of which were further evaluated for their inhibitory activities against egg hatching of *C. elegans* and J2 *M. incognita* and the wild nematode N2 L4 eggs. Only 1,4-naphthoquinone (**122**, Figure 5) appeared to be an active compound that could not only kill N2 L4 nematodes (LC_50_ 42.26 ± 2.53 μg/mL), and inhibit egg hatching of N2 (LC_50_ 34.83 ± 0.58 μg/mL), but also showed toxicity on more than 50% of *M. incognita* at a concentration of less than 50 μg/mL (LC_50_ 33.51 ± 0.21 μg/mL). The results obtained using *C. elegans* demonstrated that compound **122** could influence reactive oxygen production, superoxide dismutase activity, and the heat-shock transcription factor (HSF)-1 pathway, suggesting that compound **122** stimulated significant oxidative stress [85].

Deguelin (**123**, Figure 6) rotenone and other rotenoids, such as β-rotenolone (12aβ-hydroxyrotenone), tephrosin (12aα-hydroxydeguelin), 12a*R*-hydroxyrotenone, and dehydrorotenone, are flavonoids which were extracted from the family of Leguminosae (e.g., *Lonchocarpus*, *Derris*, *Cassia*, and *Tephrosia*) [86,87,88,89,90]. In fact, resins extracted from the roots of *Lonchocarpus utiliz* (cube), *Lonchocarpus urucu* (barbasco), *Derris elliptica* (tuba plant), or *Derris involuta* (jewel vine) were shown to contain rotenoids, and have been used in many countries as natural insecticides, acaricides, and/or piscicides [91,92]. Denguilin, when assayed against *H. contortus*, showed inhibitory activity with IC_50_ values, depending from the time of exposure (24, 48, and 72 h), of 81, 54, and 21 μM for exsheathed L3 mortality and 11.39, 25.4, and 0.004 μM for L4 mortality, respectively. Rotenoid **123** enhanced oxidative phosphorylation in mitochondria. In fact, in both parasitic (*H. contortus*) and free-living (*C. elegans*) nematodes, measurements of oxidative phosphorylation in response to deguelin (**123**, Figure 6) treatment resulted in a decrease in oxygen consumption. Thus, the toxicity of this compound could be ascribed to its ability to modulate the oxidative phosphorylation process in nematodes [93].

The acetone extract of *Heterotheca inuloides* showed strong toxicity against *Nacobbus aberrans* (Tylenchida: Pratylenchidae), which is one of the main plant-parasitic nematodes species that affects crops in Mexico, with consequent heavy economic losses. Sesquiterpenes, belonging to cadinene subgroup, such as 4-methoxyisocadalene, 7-hydroxycadalene, (4*R*)-7-hydroxy-3,4-dihydrocadalene, 1𝛼-hydroxy-1(4*H*)-isocadalen -4-one, (1*R*,4*R*)-1-hydroxy-4*H*-1,2,3,4-tetrahydrocadalen-15-oic acid, and *rac*-3,7-dihydroxy 3(4*H*)-isocadalen-4-one (**112**–**129**, Figure 6), were isolated from the extract of the dried *H. inuloides* flowers. The natural cadinenes and some hemisynthetic analogs, such as (1*S*,4*R*)-7-hydroxycalamenane, (1*S*,4*R*)-7-acetoxy-3,4-dihydrocadalene, and (1*S*,4*R*)-7-benzoiloxy-3,4-dihydrocadalene, mansonone C, 7-acetoxycadalene, 7-benzoiloxycadalene, and acetyl and benzoyl derivatives of compound **116** (**124** and **125**, Figure 6) prepared from **124**, **125**, and **126**, respectively, by conventional chemical procedures [94,95]. All the natural and hemisynthetic compounds were tested on the immobility and mortality of *Nacobbus aberrans* at the J2 stage. Compounds **125**, **126**, and their derivatives (1*S*,4*R*)-7-hydroxy calamenane, mansonone C, 7-acetoxycadalene, and (4*R*)-7-acetoxy-3,4dihydrocadalene, (1*S*,4*R*)-7-acetoxy-3,4-dihydrocadalene and (1*S*,4*R*)-7-benzoiloxy-3,4-dihydrocadalene showed after 36 h exposure LC_50_ values of 31.30, 26.30, 25.39, 21.92, 42.31, 36.19, 31.08, and 111.37 mg/L, respectively. Among all the compounds tested, nematodes were more susceptible to hydroxylated (**125**–**129**) and quinone (monsonone) compounds, whereas the remaining compounds showed moderate or no activity. The presence of the hydroxyl group seemed to be an important structural feature for nematocidal activity [96].

Suitable formulation of the ethyl acetate extract of *W. indica*, which was collected in Vietnam, reduced the formation of galls and the egg masses of *M. incognita* on the tomato roots in a dose-dependent manner. Here, 5′-Methoxywaltherione A, waltherione A waltherione C, three 4-quinolone alkaloids (**130**–**132**, Figure 6), were isolated from this extract, and exhibited strong nematocidal activity against the same organism. In particular, when assayed against *Meloidogyne arenaria*, *Meloidogyne hapla*, *M. incognita*, and *B. xylophilus*, the compound **130**–**132** at 72 h exposure, in comparison abamectin used as posto itive control, showed EC_50_ values of 0.25, 0.63, and 10.67 μg/mL; 0.09, 1.74 and 19.79 μg/mL, 0.09, 0.27 and 16.59 μg/mL, and 2.13, 3.54 and 790.85 μg/mL, respectively. Furthermore, the plant extract formulation significantly reduced gall formation on the roots of melon plants and the population density of nematodes in soil compared with the untreated control. Here, 5-Methoxywaltherione A and waltherione A (**130** and **131**) induced high mortality in the juvenile stage of all nematodes tested. The order of efficacy of the three compounds was **130** > **131** > **132**. Waltherione C (**132**) exhibited significant nematocidal activity against only root-knot nematodes [97].

Furthermore, *cis*-Dehydromatricaria ester (**133**, Figure 6) was isolated for the first time from *Tanacetum falconeri*, collected in Astore (Daosai), Pakistan. This organic extract showed nematocidal and insecticidal activity. Compound **133** showed strong nematocidal activity against *M. incognita*. The EC_50_ values recorded, at exposure times of 24, 36, and 72 h, were 3.4, 0.18, and 0.04 mg/L, respectively [98].

Additionally, 3β-Angeloyloxy-6β-hydroxyfuranoeremophil-1(10)-ene (**134**, Figure 6), the main secondary metabolite extracted from the roots of *Senecio sinuatos*, showed nematocidal activity against the second-stage juveniles (J2) of *M. incognita* and *N. aberrans*. Compound **134** was alkaline hydrolyzed to produce a derivative which, in turn, was differently esterified with anhydride acetic, benzoic acid, 2-nitrobenzoic acid, 2-bromobenzoic acid, 4-nitrobenzoic acid, 4-bromobenzoic acid, and 4-methoxybenzoic acid to produce the corresponding 6-*O*-acetyl ester and benzoyl esters. All compounds and the corresponding benzoic acids were tested for nematocidal activity against *M. incognita* and *N. aberrans* J2 using fluopyram as a positive control. In particular, the benzoyl esters possess more nematocidal activity than the corresponding free benzoic acids, while compound **134** had nematocidal activity against M. incognita when assayed at 10 μg/mL, and this effect was more nematostatic as the concentration decreased at the most effective time of 72 h [99].

## 5. Conclusions

Nematodes are one of several enemies that induce severe economic losses in agrarian production, forcing farmers to use different methods to prevent their growth and diffusion. Among these methods, the last five-to-six decades have seen a massive and extensive use of chemicals, with heavy negative consequences, such as an increase in environmental pollution and risks for human and animal health. A negative effect of the use of chemicals in agriculture is also their noteworthy contribution to climate change. The development of new control strategies based on natural products with high efficacy and selectivity has become an emergency. This review reports, for the first time, a complete overview of the microbial and plant metabolites with nematocidal activity and, thus, they are a potential applications in suitable formulations in greenhouses and fields. All the results described are summarized in Table 1. The compounds selected for their efficacy and specific nematocidal activity should be investigated firstly for their human, animal, and environmental toxicological effects. Then, for the promising compounds, a total, convenient, and ecofriendly synthesis should be developed for their large production at an industrial level.

## Figures and Tables

**Figure 1 toxins-14-00849-f001:**
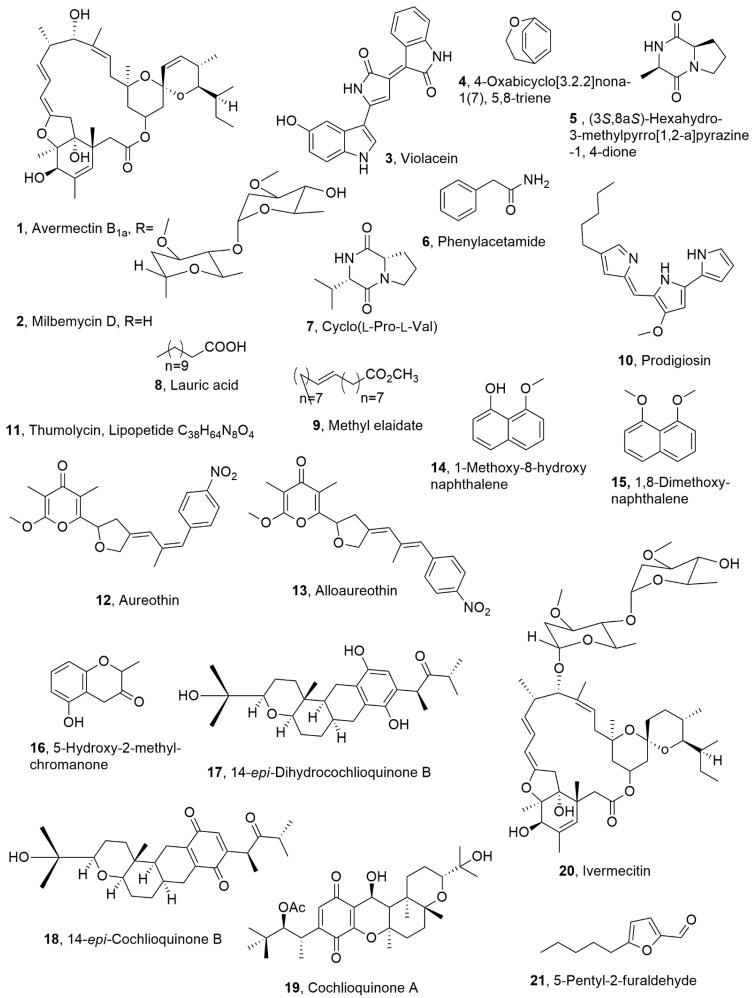
Metabolites with nematocidal activity produced by *Streptomyces avermentilis* (**1**), *Streptomyces hygroscopicus* and *Streptomyces cyaneogriseus* (**2**), several gram-negative bacteria (**3**), *Bacilus* strains (**4**–**9**), *Serriata marcescens* (**10**), *Bacillus thuringiensis* (**11**), *Streptomyces* (**12** and **13**), *Daldinia concentrica* (**14**–**16**), *Neobulgaria pura* (**17** and **18**), *Helminthosporium* sp. (**19**), *Streptomyces avermitilis* (**20**), and an unidentified species of Dermateaceae (**21**).

**Figure 2 toxins-14-00849-f002:**
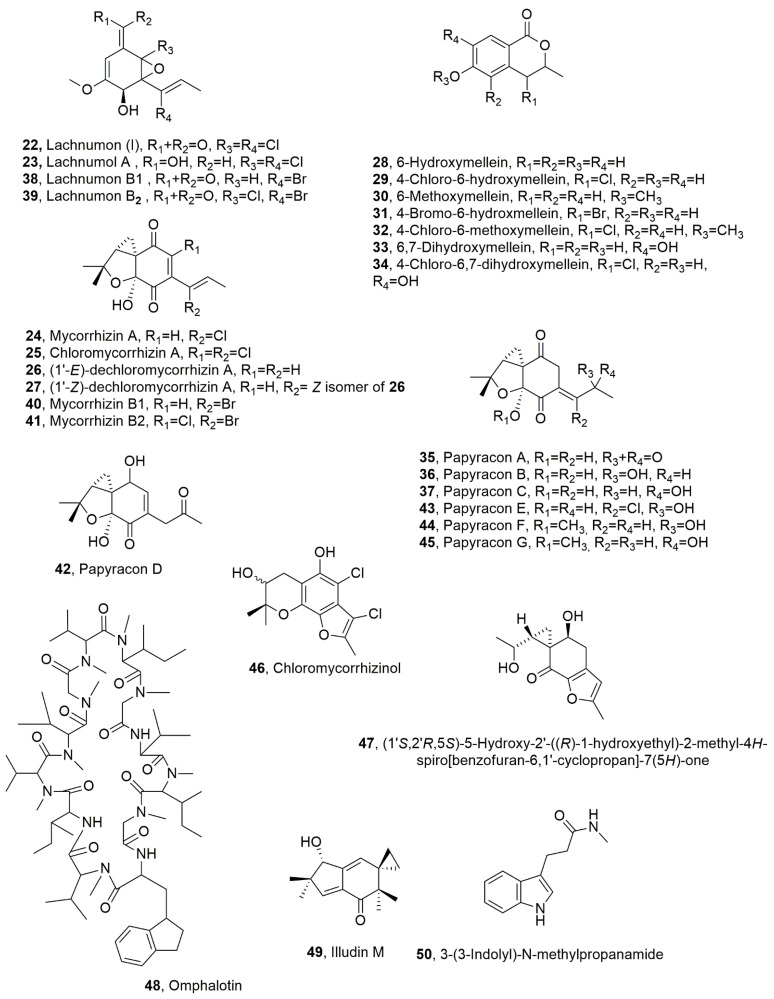
Metabolites produced by *Lachnum papyraceum* and in part also from *Gilmaniella humicola* (**22**–**34**, **35**–**37**, **38**–**41**, **42**–**45**, and **47**) and by *Omphalotus olearius (***48**–**50**).

**Figure 3 toxins-14-00849-f003:**
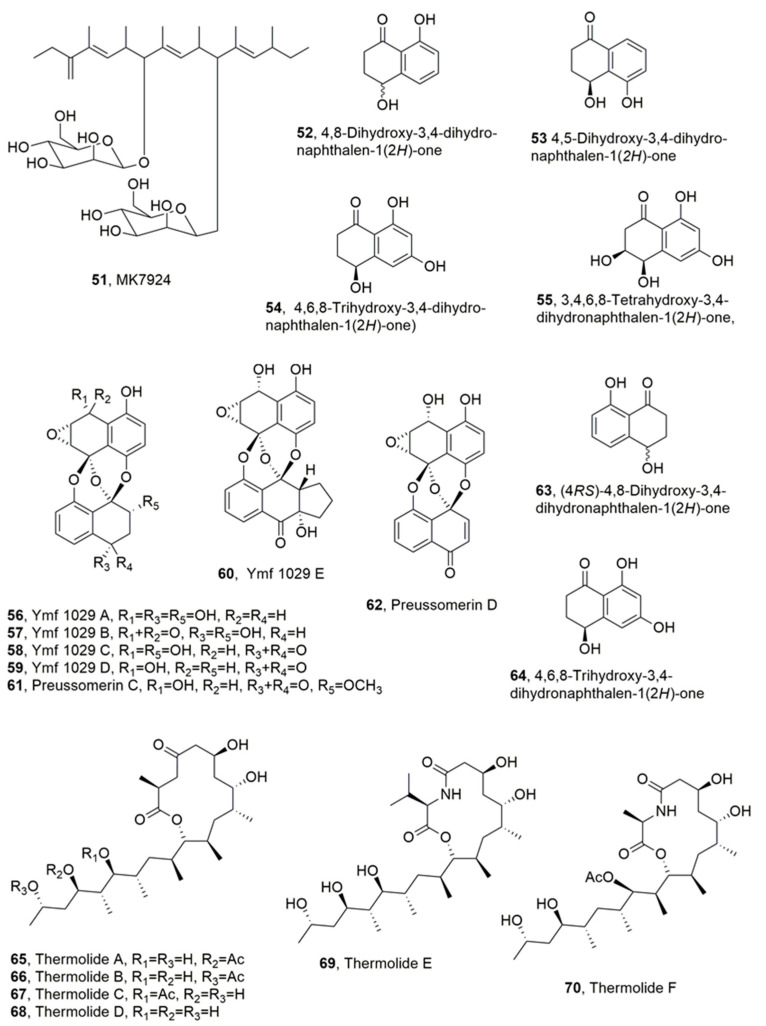
Metabolites produced by *Coronophora gregaria* (**51**), *Coronophora gregaria* (**52**–**55**), unidentified freshwater fungus YMF 1.01029 (**56**–**64**), and *Talaromyces thermophilus* (**65**–**70**).

**Figure 4 toxins-14-00849-f004:**
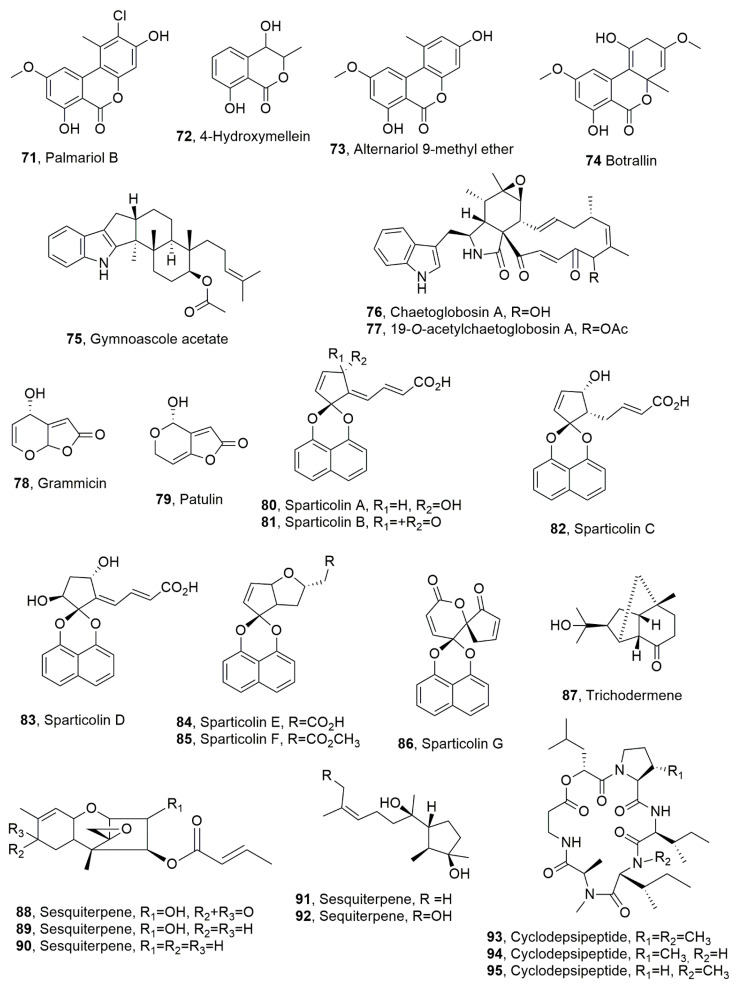
Metabolites produced by *Hyalodendriella* sp (**71**–**74**), *Gymnoascus reessii* za-130 (**75**), *Ijuhya vitellina* (**76** and **77**), *Xylaria grammica* (**78**), *Penicillium griseofulvum*, *Penicillium expansum* (**79**), a new species of Dothideomycetes (**80**–**86**), and *Trichoderma longibrachiatum* (**87**–**92** and **93**–**95**).

**Figure 5 toxins-14-00849-f005:**
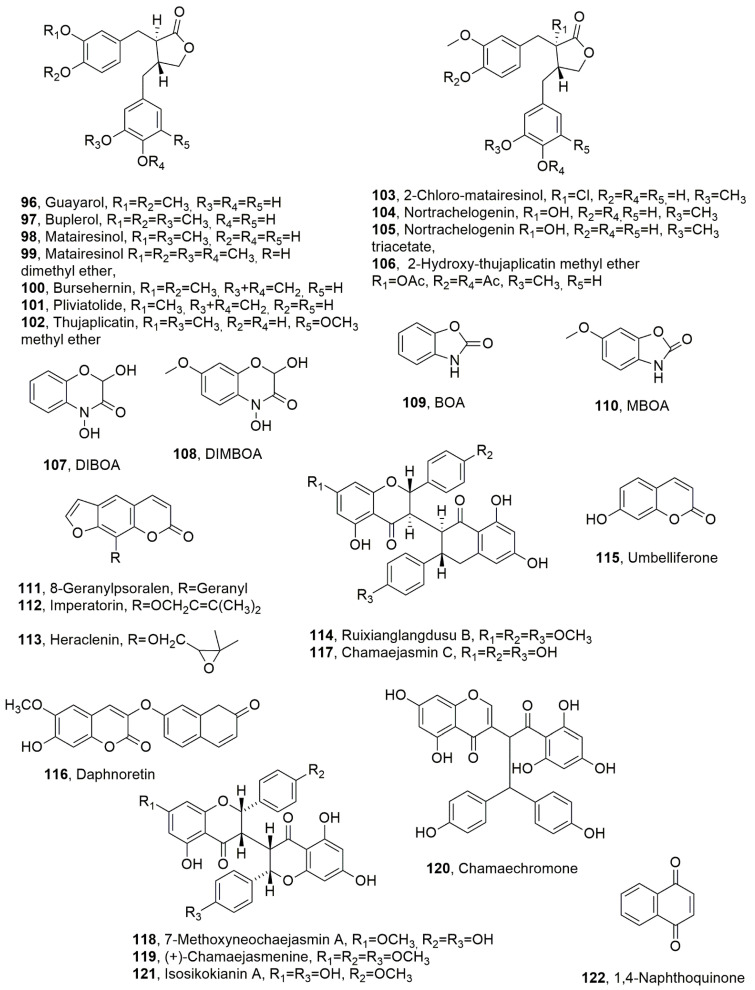
Metabolites produced by *Bupleurum salicifolium* (**96**–**102** and **103**–**106**), plants of the Poaceae family (**107**–**110**), Apiaceae, Rutaceae, Asteraceae, and Fabaceae (**111**–**113**), *Stellera chamaejasme* (**114**–**121**) and *Tagetes* spp., *Azadirachta indica*, and *Capsicum frutescens* (**122**).

**Figure 6 toxins-14-00849-f006:**
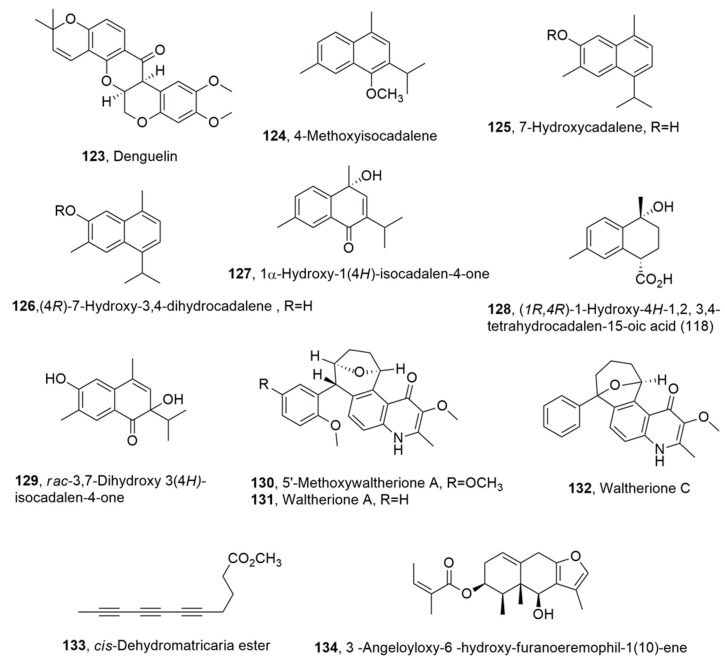
Metabolites produced by the Leguminosae family (**123**), *Heterotheca inuloides* (**124**–**129**), *Waltheria indica* (**130**–**132**), *Tanacetum falconeri* (**133**), and *Senecio sinuatos* (**134**).

**Table 1 toxins-14-00849-t001:** Microbial and plant metabolites.

Compounds	Source	Biological Activity	References
	**Metabolites Produced by Bacteria**		
Avermectin B1a (1)	*Streptomyces avermentilis*	Nematocidal activity against *Haemonchus contortus*, *Ostertagia circumcincta*; *Trichostrongylus axei*; *Trichostrongylus colubriformis*; *Cooperia* spp.; *Oesophagostomum columbianu*, (LD_50_ values in mice approximatively ranking between 15 and 50 mg/kg) and *Meloidogyne incognita or Rotylenchulus reniformis* (LD_50_/2 h 1.56 and 32.9 µg/mL , respectively)	[22]
milbemycin D (2)	*Streptomyces hygroscopicus* and *Streptomyces cyaneogriseus*	Nematocidal activity against *Caenorhabditis elegans* (EC_50_ 89 nM*)*	[22]
Violacein (3)	*Chromobacterium violaceum*, *Janthinobacterium lividum*, and *Pseudoalteromonas tunicata* D2 *Collimonas* sp., *Duganella* sp. and *Pseudoalteromonas* spp.	Antibacterial, antitrypanocidal, anti-ulcerogenic, anticancer, antioxidant, leishmanicidal, antifungal, and antiviral	[24] [27]
		Oxidative stress resistance,	[24]
		defense of amphibians from fungal disease	[25]
	*Pseudoalteromonas tunicate*	Antipredator defense mechanism against protozoan grazers	[26]
	*Microbulbifer* sp. D250	Nematocidal activity against *C. elegans* (LC_5_0 > 30 nM)	[27]
4-Oxabicyclo[3.2.2]nona-1(7), 5,8-triene (4)	*Bacillus* strain SMrs28	Nematocidal activity against *B. sxylophilus* and *D. destructor* (LD_50_ 904.12 and 1594.0 μg/mL/72 h, respectively)	[28]
(3S,8aS)-Hexahydro-3-methylpyrro[1,2-a]pyrazine-1,4-dione (5)	“	Nematocidal activity against *B. sxylophilus* and *D. destructor* (LD_50_ 451.26 and 366.62 μg/mL/72 h, respectively)	“
Phenylacetamide (6)	“	Nematocidal activity against *B. sxylophilus* and *D. destructor* (LD_50_ 232.98 and 206.38 μg/mL/72 h, respectively) “	“
Cyclo(L-Pro-L-Val) (7)	“	No tested	“
Lauric acid (8)	“	“	“
Methyl elaidate (9)	“	“	“
Prodigiosin (10)	*Serratia marcescens*	Nematocidal activity against *R. similis* and *M. javanica* (LC_50_ 83 and 79 μg/mL, respectively)	[31]
Thumolycin (11)	*Bacillus thuringiensis*	Nematocidal activity against *C. elegans* (600 U)	[32]
Aureothin (12)	*Streptomyces* sp. AE170020	Nematocidal activity against *B. xylophilus* J2s, J3s and J4s (LC_50_ 0.81, 1.15 and 1.54 μg/ L)	[34]
Alloaureothin (13)	“	Nematocidal activity against *B. xylophilus* J2s, J3s and J4s (LC_50_ 0.83, 1.10 and 1.47 μg/ L)	“
	**Metabolites produced by fungi**		
1-Methoxy-8-hydroxynaphthalene (14),	*Daldinia concentrica*	Cytotoxic, antibiotic, and nematocidal activity against *C. elegans* (LD_50_ 10 μg/mL)	[35] [36]
1,8-Dimethoxynaphthalene (15)	“	Cytotoxic, antibiotic, and nematocidal activity against *C. elegans* (LD_50_ 25 μg/mL)	“
5-Hydroxy-2-methylchromanone (16)	“	Nematocidal activity *C. elegans* (LD_50_ > 100 μg/mL)	[36]
14-*epi*-Dihydrocochlioquinone B (17)	*Neobulgaria pura*	Nematocidal against *C. elegans* (LD_50_ 10 μg/mL) and *M. incognita* (LD_30_ 100 μg/mL)	[35,37]
14-*epi*-Cochlioquinone B (18)	“	Nematocidal activity against *C. elegans* and *M. incognita*	“
Cochlioquinone A (19)	*Helminthosporium* sp.	“	[38]
Ivermectin (20)	*Streptomyces avermitilis*	“	[39]
5-Pentyl-2-furaldehyde (21)	Dermateaceae sp. *Irpex lacteus*	Nematocidal activity against *C. elegans* (LD_50_ 60 μg/mL) and *M. incognita* (LD_50_ 75 μg/mL)	[40]
Lachnumon (I) (22)	*Lachnum papyraceum*	Nematocidal activity against *C. elegans* (LD_50_ 25 μg/mL) and *M. incognita* (LD_50_ > 100μg/mL)	[41,42]
Lachnumol A (23)	“	Nematocidal activity against *C. elegans* (LD_50_ 5 μg/mL) and *M.incognita* (LD_50_ 100 μg/mL)	“
Mycorrhizin A (24),	“	Nematocidal activity against *C. elegans* (LD_50_ 1 μg/mL) and *M. incognita* (LD_50_ 100 μg/mL)	“
Chloromycorrhizin A (25)	“	Nematocidal activity against *C. elegans* (LD_50_ 100 μg/mL) and *M. incognita* (LD_50_ 100 μg/mL)	“
(1’-*E*)-dechloromycorrhizin A (26)	“	Nematocidal activity against *C. elegans* (LD_50_ 2 μg/mL) and *M. incognita* (LD_50_ > 100μg/mL)	“
(1’-*Z*)-dechloromycorrhizin A (27)	“	Nematocidal activity against *C. elegans* (LD_50_ 2 μg/mL) and *M. incognita* (LD_50_ > 100 μg/mL)	“
6-Hydroxymellein (28)	“	No nematocidal activity	[35]
4-Chloro-6-hydroxymellein (29)	“	“	“
6-Methoxymellein (30)	“	“	“
4-Bromo-6-hydroxmellein (31)	“	“	“
4-Chloro-6-methoxymellein (23)	“	“	“
6,7-dihydroxymellein (33)	“	*”*	“
4-Chloro-6,7-dihydroxymellein (34)	“	Nematocidal activity against *C. elegans*	“
Papyracon A (35)	“	Moderate nematocidal activity against *C. elegans* (LD_50_ 25 μg/mL) and *M. incognita* (LD_50_ > 100 μg/mL)	“
Papyracon B (36)	“	Moderate nematocidal activity against *C. elegans* (LD_50_ 50 μg/mL) and *M. incognita* (LD50 > 100 μg/mL)	“
Papyracon C (37)	“	Moderate nematocidal activity against *C. elegans* (LD_50_ 50 μg/mL) and *M. incognita* (LD50 > 100 μg/mL)	“
Lachnumon B1 (38)	“	Nematocidal activity against *C. elegans* (LD_50_ 50 μg/mL) and *M. incognita* (LD50 > 100 μg/mL)	“
Lachnumon B2 (39)	“	Moderate nematocidal activity against *C. elegans* (LD_50_ 25 μg/mL) *and M. incognita* (Not tested)	“
Mycorrhizin B1 (40)	“	Moderate nematocidal activity against *C. elegans* (LD_50_ 2 μg/mL) and *M. incognita* (LD_50_ > 100 μg/mL)	“
Mycorrhizin B2 (41)	“	Moderate nematocidal activity against *C. elegans* (LD_50_ 5 μg/mL) and *M. incognita* (LD_50_ > 100 μg/mL)	“
Papyracon D (42)	“	Moderate nematocidal activity against *C. elegans* (LD_50_ 50 μg/mL) and *M. incognita* (LD_50_ > 100 μg/mL)	“
Papyracon E (43)	“	Moderate nematocidal activity against *C. elegans* (LD_50_ 50 μg/mL) and *M. incognita* (LD_50_ > 100 μg/mL)	“
Papyracon F (44)	“	Moderate nematocidal activity against *C. elegans* (LD_50_ 50 μg/mL) and *M. incognita* (LD_50_ > 100 μg/mL)	“
Papyracon G (45)	“	Moderate nematocidal activity against *C. elegans* (LD_50_ 50 μg/mL) and *M. incognita* (LD_50_ > 100 μg/mL)	“
Chloromycorrhizinol (46)	“	Nematocidal activity against *C. elegans* (LD_50_ 100 μg/mL) and *M. incognita* (LD_50_ > 100 μg/mL)	“
(*1’S,2’R,5S*)-5-Hydroxy-2’-((*R*)-1-hydroxyethyl)-2-methyl-4*H*-spiro[benzofuran-6,1’-cyclopropan]-7(5*H*)-one (47)	“	Not tested	“
Omphalotin (48)	*Omphalotus olearius*	Nematocidal activity against *C. elegans* (LD_50_ 0.57 μg/mL) and *M. incognita* (LD_50_ 18.95 μg/mL)	[48].
Illudin M the (49)	*“*	No activity	“
3-(3-Indolyl)-N-methylpropanamide (50)	*“*	“	“
MK7924 (51)	*Coronophora gregaria*	Nematocidal activity against *C. elegans* at 100 μg/mL	[55]
4,8-Dihydroxy-3,4-dihydronaphthalen-1(2*H*)-one (52),	*Caryospora callicarpa*	Nematocidal activity against *B. xylophilus* (LD_50_ 540.2, 436.6 and 209.0 at 12, 24, and 36 h, respectively)	[56]
4,5-Dihydroxy-3,4-dihydronaphthalen-1(2*H*)-one (53)	“	Nematocidal activity against *B. xylophilus* (LD_50_ 1169.8, 461.3, and 229.6 at 12, 24, and 36 h)	“
4,6,8-Trihydroxy-3,4-dihydronaphthalen-1(2*H*)-one (54)	“	Nematocidal activity against *B. xylophilus* (LD_50_ 1011.6, 522.5, and 220.3 at 12, 24, and 36 h)	“
3,4,6,8-Tetrahydroxy-3,4-dihydronaphthalen-1(2*H*)-one (55)	“	Nematocidal activity against *B. xylophilus* (LD_50_ 854, 468, and 206.1 at 12, 24 and 36 h)	“
Ymf 1029 A (56)	Unidentified freshwater fungus YMF 1.01029	Nematocidal activity against *B. xylophilus*	[60]
Ymf 1029 B (57)	“	“	“
Ymf 1029 C (58)	“	“	“
Ymf 1029 D (59)	“	“	“
Ymf 1029 E (60)	“	“	“
Preussomerin C (61)	“	“	“
Preussomerin D (62)	“	“	“
(4*RS*)-4,8-dihydroxy-3,4-dihydronaphthalen-1(2*H*)-one (63)	“	“	“
4,6,8-Trihydroxy-3,4-dihydronaphthalen-1(2*H*)-one (64)	“	“	“
Thermolide A (65)	*Talaromyces thermophilus*	Nematocidal activity against *M. incognita*, *B. siylopilus*, and *P. redivevus*	[62]
Thermolide B (66)	“	“	“
Thermolide C (67)	“	“	“
Thermolide D (68)	“	“	“
Thermolide E (69)	“	Not tested	“
Thermolide F (70)	“	“	“
Palmariol B (71)	*Hyalodendriella* sp.	Antimicrobial, inhibition of acetylcholinesterase, and nematocidal activity against *C. elegans* (IC_50_ 56.21 μg/mL)	[64]
4-Hydroxymellein (72)	“	Antibacterial and nematocidal activity against *C. elegans* (IC_50_ 86.86 μg/mL)	“
Alternariol 9-methyl ether (73)		Nematocidal activity against *C. elegans* (IC_50_ 93.99 μg/mL)	“
Botrallin (74)	“	Nematocidal activity against *C. elegans* (IC_50_ 84.51 μg/mL)	“
Gymnoascole acetate (75))	*Gymnoascus reessii* za-130	Nematocidal activity against *M. incognita* J2s (EC_50_ 47.5 μg/mL)	[65]
Chaetoglobosin A (76)	*Ijuhya vitellina*	Nematocidal activity against *H. filipjevi* (at 50 μg/mL, it paralyzed the nematode)	[66]
19-*O*-Acetylchaetoglobosin A (77)	“	Nematocidal activity against *H. filipjevi* (at 100 μg/mL, it paralyzed the nematode)	“
Grammicin (78)	*Xylaria grammica*	Low antibacterial and cytotoxic activity, but nematocidal activity against *M. incognita* J2 juvenile mortality and egg-hatching inhibition (EC_50_ 15.95 and 5.87 μg/mL)	[67]
Patulin (79)	*Penicillium griseofulvum* and *Penicillium expansum*, and other fungal genera	Antibacterial, cytotoxic, and dematocidal activity against *M. incognita* J2 juvenile mortality and eggs-hatching inhibition (EC_50_/72 h 115.67 μg/mL)	[67,69,70]
Sparticolin A (80)	*Sparticola junci*	Nematocidal activity against *C. elegans* (LD_50_ 50 μg/mL)	[73]
Sparticolin B (81)	“	Nematocidal activity against *C. elegans* (LD_50_ 50 μg/mL)	“
Sparticolin C (82)	“	Nematocidal activity against *C. elegans* (LD_50_ 25 μg/mL)	“
Sparticolin D (83)	“	Nematocidal activity against *C. elegans* (LD_50_ 50 μg/mL)	“
Sparticolin E (84)	“	Nematocidal activity against *C. elegans* (LD_50_ 50 μg/mL)	“
Sparticolin F (85)	“	Nematocidal activity against *C. elegans* (LD_50_ 12.5 μg/mL)	“
Sparticolin G (86)	“	Antifungal and cytotoxic activity, but not tested for nematocidal activity	“
Trichodermene (87)	*Trichoderma longibrachiatum*	Antifungal activity against *Colletotrichum lagenarium*	[74]
Sesquiterpene (88)	“	“	“
Sesquiterpene (89)	“	“	“
Sesquiterpene (90)	“	No activity	“
Sesquiterpene (91)	“	No activity	“
Sesquiterpene (92)	“	No activity	“
Cyclodepsipeptide (93)	“	Nematocidal against *M. incognita* (IC_50_ 149.2 μg/mL)	“
Cyclodepsipeptide (94)	“	Nematocidal against *M. incognita* (IC_50_ 140.6 μg/mL)	“
Cyclodepsipeptide (95)	“	Nematocidal against *M. incognita* (IC_50_ 198.7 μg/mL)	“
**Metabolites produced by plants**			
Guayarol (96)	*Bupleurum salicifolium*	Nematocidal activity against *G. pallida* and *G. rostochiensis* (LC_50_ range 2 × 10^−6^ to 1.26 × 10^−3^)	[80]
Buplerol (97)	*“*	“	“
Matairesinol (98)	*“*	Nematocidal activity against *G. pallida* and *G. rostochiensis* (70% hatching reduction)	“
Matairesinoldimethyl ether (99)	*“*	Nematocidal activity against *G. pallida* and *G. rostochiensis* (LC_50_ range 2 × 10^−6^ to 1.26 × 10^−3^)	“
Bursehernin (100)	*“*	Nematocidal activity against *G. pallida* and *G. rostochiensis* (550% hatching reduction; HID 16.42)	“
Pliviatolide (101)	*“*	Nematocidal activity against *G. pallida* and *G. rostochiensis* (LC50 range 2 × 10^−6^ to 1.26 × 10^−3^)	“
Thujaplicatin, methyl ether (102)	*“*	“	“
2-Chloro-matairesinol (103)	*“*	“	“
Nortrachelogenin (104)	*“*	“	“
Nortrachelogenin triacetate (105)	*“*	“	“
2-Hydroxythujaplican methyl ether (106)		“	“
DIBOA (107)	Plant of Poaceae family	Nematocidal activity against *M. incognita* (LD_50_/168 h 74.3 μg/mL; J2 mortality LD_50_ 20.9 μg/mL) and *X. americanum* (LD_50_/24 h 18.4 μg/mL)	[81]
DIMBOA (108)	“	Nematocidal activity against *M. incognita* J2 mortality (LD _50_ 46.1 μg/mL) and *X. americanum* (LD_50_/24 h 48.3 μg/mL)	“
BOA (109)	“	No activity	“
MBOA (110)		Nematocidal activity against *M. incognita* J2 mortality (LD_50_ 49.2 μg/mL)	“
8-Geranylpsoralen, (111)	Apiaceae, Rutacea, Asteraceae and Fabaceae plant families	Nematocidal activity against *B. xylophilus* LD_50_/72 h 188.3 μg/mL) and *P. redivivus* (LD_50_/72 h 117.5 μg/mL)	[82]
Imperatorin (112)	“	Nematocidal activity against *B. xylophilus* LD_50_/72 h 161.7 μg/mL) and *P. redivivus* (LD_50_/72 h 179.0 μg/mL)	“
Heraclenin (113)	“	Nematocidal activity against *B. xylophilus* LD_50_/72 h 114.7 μg/mL) and *P. redivivus* (LD_50_/72 h 184.7 μg/mL)	“
Ruixianglangdusu B (114)	*Stellera chamaejasme* L.	Nematocidal activity against *B. xylophilus* (LC_50_ at 12, 24, and 72 h exposure at 227.4, 71.6, and 15.7 μM, respectively) and *B. mucronatus* (LC_50_ at 12, 24, and 72 h exposure at 1.8 × 10^3^, 160.2, and 0.6 μM, respectively)	[83]
Umbelliferone (115)	*“*	Nematocidal activity against *B. xylophilus* (LC_50_ at 12, 24, and 72 h exposure of 1.3 × 10^7^, 5.7 × 10^7^, and 3.3 μM, respectively) and *B. mucronatus* (LC_50_ at 12, 24, and 72 h exposure of 2.6 × 10^3^, 851, and 33.4 μM, respectively)	“
Daphnoretin (116)	*“*	Nematocidal activity against *B. xylophilus* (LC_50_ at 12, 24, and 72 h exposure 47.8, 3.1, and 2.7 μM, respectively) and *B. mucronatus* (LC_50_ at 12, 24, and 72 h exposure of 2.3 × 10^6^, 169.9, and 3.1 μM, respectively)	“
Chamaejasmenine C (117)	*“*	Nematocidal activity against *B. xylophilus* (LC_50_ at 12, 24, and 72 h exposure of 1.7 × 10^4^, 1.1 × 10^4^, and 65.3 μM) and *B. mucronatus* (LC_50_ at 12, 24, and 72 h exposure of 463.5, 156.7, and 0.05 μM, respectively)	“
7-Methoxyneochaejasmin A (118)	*“*	Nematocidal activity against *B. xylophilus* (LC_50_ at 12, 24 and 72 h exposure <0.001, 3.4 and 167.3 μM) and *B. mucronatus* (LC_50_ at 12, 24 and 72 h exposure 1.8 × 10^4^, 384.2, and 151.1 μM)	“
(+)-Chamaejasmine (119)	*“*	Nematocidal activity against *B. xylophilus* (LC_50_ at 12, 24, and 72 h exposure of 16.5, 8.8, and 4.7 μM, respectively) and *B. mucronatus* (LC_50_ at 12, 24, and 72 h exposure of 1.8 × 10^3^, 1.6 × 10^3^, and 5.1 × 10^3^ μM, respectively)	“
Chamaechromone (120)	*“*	Nematocidal activity against *B. xylophilus* (LC_50_ at 12, 24, and 72 h exposure of 0.7, 10.3, and 36.7 μM, respectively) and *B. mucronatus* (LC_50_ at 12, 24, and 72 h exposure of 327, 5.7, and 0.003 μM, respectively)	“
Isosikokianin A (121)	*“*	Nematocidal activity against *B. xylophilus* (LC_50_ at 12, 24, and 72 h exposure of 147.7, 385.2, and 2.2 × 10^2^ μM, respectively) and *B. mucronatus* (LC_50_ at 12, 24, and 72 h exposure of 2.6 × 10^4^, 32.5, and 2.3 μM, respectively)	
1,4-Naphthoquinone (122)	*Rubia wallichiana*	Nematocidal activity against N2L4 (kill LC_50_ value 42.26 μg/mL), N2 (inhibition of egg hatching LC_50_ 34.83 μg/mL), *M. incognita* (LC_50_ 33.51 μg/mL)	[85]
Denguelin (123)	Leguminosae spp.	Nematocidal activity against *H. contortus* (L3 mortality IC_50_/24,48,72 h 81, 54 and 21 μM) (L4 mortality IC_50_/24,48,72 h 11.39, 25.4 and 0.004 μM)	[93]
4-Methoxyisocadalene (124)	*Heterotheca* inuloides	No activity	[96]
7-Hydroxycadalene (125)	*“*	Nematocidal activity against *N. aberrans* (mortality at J2 stage (LC_50_ 31.30 mg/mL)	“
(*4R*)-7-hydroxy-3,4-dihydrocadalene (126)	*“*	Nematocidal activity against *N. aberrans* (mortality at J2 stage LC_50_ 26.30 mg/mL)	“
1𝛼-Hydroxy-1(4*H*)-isocadalen-4-one (127)	*“*	No activity	“
(*1R,4R*)-1-Hydroxy-4*H*-1,2, 3,4-tetrahydrocadalen-15-oic acid (128)	*“*	“	“
*rac*-3,7-Dihydroxy 3(4*H*)-isocadalen-4-one (129)	*“*	“	
5′-Methoxywaltherione A (130)	*Waltheria indica*	Nematocidal activity against *M. arenaria* (EC_50_ 0.25 μg/mL), *M. hapla* (E_50_ 0.09 μg/mL), *M. incognita* a(E_50_ 0.09 μg/mL), and *B. xylophilus* (E_50_ 2.13 μg/mL)	[97]
Waltherione A (131)	*“*	Nematocidal activity against *M. arenaria* (EC_50_ 0.63 μg/mL), *M. hapla* (E_50_ 1.74 μg/mL), *M. incognita* a(E_50_ 0.27 μg/mL), and *B. xylophilus* (E_50_ 3.54 μg/mL)	“
Waltherione C (132)	*“*	Nematocidal activity against *M. arenaria* (EC_50_ 10.67 μg/mL), *M. hapla* (E_50_ 19.79 μg/mL), *M. incognita* (E_50_ 16.59 μg/mL), and *B. xylophilus* (EC_50_ 790.85 μg/mL)	“
cis-Dehydromatricaria ester (133)	*Tanacetum falconeri*	Nematocidal activity against *M. incognita* (EC_50_/24,36, 72 h 3.4, 0.18, and 0.04 mg/L)	[98]
3β-Angeloyloxy-6β-hydroxyfuranoeremophil-1(10)-ene (134)	*Senecio sinuatos*	Nematocidal activity against *M. incognita* (at 10 mg/mL)	[99]

**“** Means the same content.

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
