# Peer review of "Microbial and Plant Derived Low Risk Pesticides Having Nematocidal Activity"

_toxins, 2022, doi:10.3390/toxins14120849_

Round 1

Reviewer 1 Report

1. The title "Microbial and plant and metabolites with nematocidal activity" need to be corrected.
2. The manuscript's content is adequate, but extensive editing of the English language and style is required throughout. There are numerous ambiguous sentences.
3. The sentence "Among all the enemies that induces severe economic losses to the agrarian production there are, microbial pathogens, virus, weeds including parasitic plants, insects and nematodes." is not clear.
4. Spelling mistakes e.g. "suvervival" line 33. There are several typographical, grammatical, and spelling mistakes throughout the manuscript.

5. Add some recent reference in the manuscript.

Author Response

see file attached

Reviewer 2 Report

This review is focused on the microbial and plant metabolites with nematocidal activity with potential application in suitable formulation in green-house and field. The manuscript is well-written and is a good contribution to the agricultural field to combat the natural enemies like nematodes. However, I have some comments about the manuscript:

1. What are the contents of these compounds with potential nematocidal activity in microbial or plant metabolites? Are these metabolites easy to produce and prepare?

2. How about the stability of these compounds?

3. Page 18 Table 1. The caption of this table is not proper. Not all the compounds in the table have nematocidal activity. It is better to add the LC50 values in the table. What are the definitions of strong or moderate activity?

4. The authors should talk about future study directions and applications about these natural compounds at the end of the manuscript.

Author Response

see file attache

Reviewer 3 Report

Dear Authors, 

The review entitled "Microbial and plant and metabolites with nematocidal activity" has been submitted to Toxins journal in Plant Toxins section. 

A huge effort was made to write this review dealing with microbial and plant metabolites with nematocidal activity with potential application in suitable formulation in green-house and field. The results discussed in the different sections were obtained from Sci-Finder research since 1995 and chronologically reported in each paragraph. There are 98 references cited in the text from 1995 up to 2022. 

Nematodes are one of the several enemies that induces severe economic losses to the agrarian production that forced the farmers to use different methods to avoid they growth and diffusion. Among them in the last five-six decades a massive and extensive use of chemical was done determining heavy negative consequences as the increase of environmental pollution and risks for human and animal health. A negative effect of the use of chemical in agriculture is also their notheworthy contribute to climate changes The development of new control strategies based on natural products with high efficacy and selectivity is became an emergency. 

Although this piece of work is undoubtedly worth reading I got confused with its layout. There is too much information presented in only 3 sections-bacterial, fungal and plant metabolites. It is very inconvinient to find information of interest. Maybe a good idea would be to add some subsections? 

Extensive language and style editing is required.  Also please have a look for a better title. 

I would be happy to revise it again after corrections. 

Author Response

see file attached

Round 2

Reviewer 1 Report

1. List of abbreviation should be added in the manuscript. 

2. What are LD50 and LC50? It should be defined.

Reviewer 2 Report

The authors have addressed to my comments carefully. Tha manuscript can bee accepted after revising some spelling mistakes., e.g., line 136 there should be a space between which and is .

Reviewer 3 Report

Dear Authors, 

I received the revised form of the manuscript. I can see a progress in the points I have mentioned before, although there are still some issues I have to stress out.

I do not think the title is appropriate. Maybe please consider using "low risk pesticides" in it? Like "Microbial and plant derived low risk pesticides having nematocidal activity"? 

English style is much better, a lot of mistakes were corrected but the text still needs to be doublechecked. 

I will accept it after minor corrections. 
